# Effect of Self-Oscillation on Escape Dynamics of Classical and Quantum Open Systems

**DOI:** 10.3390/e22080839

**Published:** 2020-07-30

**Authors:** Minggen Li, Jingdong Bao

**Affiliations:** Department of Physics, Beijing Normal University, Beijing 100875, China; 201821140026@mail.bnu.edu.cn

**Keywords:** self-oscillation, escape dynamics, open systems, quantum fluctuation

## Abstract

We study the effect of self-oscillation on the escape dynamics of classical and quantum open systems by employing the system-plus-environment-plus-interaction model. For a damped free particle (system) with memory kernel function expressed by Zwanzig (J. Stat. Phys. 9, 215 (1973)), which is originated from a harmonic oscillator bath (environment) of Debye type with cut-off frequency $w_{d}$, ergodicity breakdown is found because the velocity autocorrelation function oscillates in cosine function for asymptotic time. The steady escape rate of such a self-oscillated system from a metastable potential exhibits nonmonotonic dependence on $w_{d}$, which denotes that there is an optimal cut-off frequency makes it maximal. Comparing results in classical and quantum regimes, the steady escape rate of a quantum open system reduces to a classical one with $w_{d}$ decreasing gradually, and quantum fluctuation indeed enhances the steady escape rate. The effect of a finite number of uncoupled harmonic oscillators *N* on the escape dynamics of a classical open system is also discussed.

## 1. Introduction

The study of open systems, which may trace back to the pioneering studies on Brownian motion [1,2], has been an important area in both classical and quantum statistical mechanics [3,4]. In the theory of open systems, the deterministic dynamics of particles in the system is replaced in the quantum regime by a stochastic Schrödinger equation, corresponding to a stochastic Langevin equation [5] in the classical limit. In the classical regime, many studies have been made for open systems by using the Caldeira–Leggett (CL) model [6,7,8] (system-plus-environment-plus-interaction model), in which the environment is often regarded as a heat bath consisted of a large set of independent harmonic oscillators. In the quantum regime, a model quantum system coupled to its environment forms the standard paradigm of quantum Brownian motion. However, the size of environment is small for systems of interest in many contexts, in particular, in mesoscopic physics and nanotechnology [9,10,11,12,13]. The thermodynamic limit may no longer be justified. There is a natural infrared cut-off for the frequency of oscillators schematizing the environment. A finite number of oscillators of a realistic heat bath is also objective. As escape of a particle from a metastable potential plays a central role in different fields of science, including condensed matter physics [14], polymer physics [15,16], and neuroscience [17], two effects on escape dynamics [18] of an open system indeed need to be considered: a finite bandwidth for the frequency of oscillators and a finite number of oscillators in a heat bath.

The aim of this paper is to analyze the effect of self-oscillation [19,20,21] caused by a finite bandwidth [22,23] and a finite number of oscillators [13,24] in a heat bath on escape dynamics of classical and quantum open systems. First, the effect of a finite bandwidth on the escape dynamics of classical and quantum systems is studied in the limit N→∞. One systematic approach is based on the Zwanzig–Mori projection operator formalism, which leads to a generalized Langevin equation (GLE) for classical open systems. Based on an initial coherent state representations of bath oscillators and an equilibrium canonical distribution of quantum mechanical mean values of their coordinates and momenta, a quantum generalized Langevin equation (QGLE) in c numbers can also be derived for quantum open systems [14,25,26]. We employ the memory kernel expressed by Zwanzig [27], i.e., $\gamma \left(t\right)=\frac{3\gamma_{0}^{2}}{w_{d}^{2}}\frac{sin(w_{d}t)}{t}$, where $w_{d}$ is a cut-off frequency and $\gamma_{0}$ constant, which is originated from the Debye cut-off for the frequency of oscillators in a heat bath. Ergodicity breakdown is found for a damped free particle as the velocity autocorrelation function (VAF) is shown to oscillate in cosine function for asymptotic time. The steady escape rate of such a self-oscillated system depends non-monotonically on $w_{d}$, which is analyzed from the perspective of two timescales: the correlation time of fluctuations, $\tau_{c}$, and the escape time, $\tau_{e}$, [28] and the change of the friction exerting on the system [29]. Comparing results in classical and quantum regimes, quantum fluctuation enhances the steady escape rate. Second, the effect of a finite number of oscillators on the escape dynamics of a classical open system is investigated by numerical simulation of (2N+2) Hamilton equations. The dependence of the steady escape rate on *N* is presented here. The effect of self-oscillation caused by many frequencies missing in the interval of interest in a finite bath on escape dynamics is also analyzed.

The paper is organized as follows. In Section 2, we present a general analysis of VAF of a damped free particle and obtain an exact analytical expression for asymptotic time. In Section 3, on the one hand, the effect of a finite bandwidth for the frequency of oscillators on escape dynamics is investigated by the numerical simulation of a GLE and a QGLE in c numbers. In particular, we analyze the nonmonotonic dependence of the steady escape rate on $w_{d}$ from the view of two timescales and a crossover between weak and strong friction regime. We also compare results of the steady escape rate in quantum and classical regimes here. On the other hand, the effect of a finite number of oscillators on escape dynamics of a classical open system is studied. The conclusions are drawn in Section 4.

## 2. General Analysis of VAF: Damped Free Particle

In the classical regime, the starting point for our analysis is the Hamiltonian of a particle plus environment consisted of N-independent harmonic oscillators [6,7,8], which means the interaction of the particles in heat bath of each other [5] has not been considered here,
(1)$$\begin{array}{c} H=\frac{P^{2}}{2M}+U\left(X\right)+\sum_{i=1}^{N}\left[\frac{p_{i}^{2}}{2m_{i}}+\frac{m_{i}w_{i}^{2}}{2}{\left(x_{i}-\frac{c_{i}X}{m_{i}w_{i}^{2}}\right)}^{2}\right]. \end{array}$$
where {X,P} and $\left\{x_{i},p_{i}\right\}$ (i=1, 2 , … , *N*) are, respectively, the test particle and the *i*th oscillator coordinates and momentums. *M* and $m_{i}$ are, respectively, the mass of the test particle and the *i*th oscillator. $w_{i}$ denotes the vibrational frequency of the *i*th oscillator. The coupling parameter $c_{i}$ characterizes the strength of the system–environment interaction. U(x) is an external potential. By writing the Hamilton equations and solving for the particles of the heat bath, a GLE is obtained:(2)$$\begin{array}{c} M\dot{V}=-U^{\prime }\left(X\right)-M\int_{0}^{t}\gamma \left(t-s\right)V\left(s\right)ds+F\left(t\right), \end{array}$$
where γ(t) denotes the memory kernel function. The noise F(t) has zero means, which satisfies the fluctuation–dissipation theorem (FDT), written as $C_{F}\left(t\right)=\langle F\left(t\right)F\left(0\right)\rangle =Mk_{B}T\gamma \left(t\right)$; here, $k_{B}$ denotes Boltzmann’s constant and T the temperature.

In the quantum regime, the total system-bath Hamiltonian can be written as [25,26,30]
(3)$$\begin{array}{c} H=\frac{{\hat{p}}^{2}}{2M}+U\left(\hat{x}\right)+\sum_{i=1}^{N}\left[\frac{{\hat{p}}_{i}^{2}}{2m_{i}}+\frac{m_{i}w_{i}^{2}}{2}{\left({\hat{x}}_{i}-\frac{c_{i}\hat{x}}{m_{i}w_{i}^{2}}\right)}^{2}\right], \end{array}$$
where $\hat{x}$ and $\hat{p}$ are the coordinate and momentum operators of the system, respectively, and $\left\{\hat{x_{i}},\hat{p_{i}}\right\}$ are the set of coordinate and momentum operators of the bath oscillators. The coordinate and the momentum operators follow the commutation relation $[\hat{x},\hat{p}]=i\hbar$ and $\left[\hat{x_{j}},\hat{p_{k}}\right]=i\hbar \delta_{jk}$. Using the Heisenberg equation of motion for operator, a QGLE in c numbers can be obtained based on an initial coherent state representations of bath oscillators and an equilibrium canonical distribution of quantum mechanical mean values of their coordinates and momenta:(4)$$\begin{array}{c} M\dot{V}+M\int_{0}^{t}\gamma \left(t-s\right)V\left(s\right)ds+U^{\prime }\left(X\right)=F\left(t\right)+Q\left(X,t\right), \end{array}$$
where V(t) and X(t) denote quantum mechanical mean values of velocity and position, respectively, expressed as $\langle \hat{v}\left(t\right)\rangle =V\left(t\right)$ and $\langle \hat{x}\left(t\right)\rangle =X\left(t\right)$. γ(t) denotes the dissipative memory kernel, given by $\gamma \left(t\right)=\int_{0}^{\infty }\frac{J(w)}{w}cos\left(wt\right)dw$, where $J\left(w\right)=\frac{1}{M}\sum_{i=1}^{N}\frac{c_{i}^{2}}{m_{i}w_{i}}\delta \left(w-w_{i}\right)$ denotes the bath spectral density. F(t) is zero centered stationary noise, i.e., 〈F(t)〉=0 and 〈F(t)F(0)〉=C(t), where C(t) is written as $C\left(t\right)=\frac{\hbar }{2}\int_{0}^{\infty }J\left(w\right)cos\left(wt\right)\times coth\left(\frac{\hbar w}{2k_{B}T}\right)dw$. Q(X,t) is a quantum fluctuation term, expressed as $Q\left(X,t\right)=U^{\prime }\left(\langle \hat{x}\rangle \right)-\langle U^{\prime }\left(\hat{x}\right)\rangle$.

For a damped free particle in both regimes, we can obtain a differential equation for the VAF $C_{V}\left(t\right)=\frac{\langle V(t)V(0)\rangle }{\langle V^{2}\left(0\right)\rangle }$ by multiplying the initial velocity V(0) and performing an ensemble average, specifically,
(5)$$\begin{array}{c} \frac{dC_{V}\left(t\right)}{dt}=-\int_{0}^{t}\gamma \left(t-s\right)C_{V}\left(s\right)ds. \end{array}$$

We employ the memory kernel expressed by Zwanzig [27], i.e.,
(6)$$\begin{array}{c} \gamma \left(t\right)=\frac{3\gamma_{0}^{2}}{w_{d}^{2}}\frac{sin(w_{d}t)}{t}, \end{array}$$
where $w_{d}$ is a cut-off frequency and $\gamma_{0}$ a constant. The parameter $\gamma_{0}=1$ is fixed. In the limit of N→∞, the frequency distribution of the oscillators can be treated as continuous with the Debye type, and the memory kernel can be obtained by setting $c_{i}=\gamma_{0}/\sqrt{N}$. The Laplace transform of the VAF reads [31] ${\tilde{C}}_{V}\left(z\right)=\frac{1}{z+\tilde{\gamma }\left(z\right)}$, where the Laplace transform of the memory kernel is given by
(7)$$\begin{array}{c} \tilde{\gamma }\left(z\right)=\frac{3\gamma_{0}^{2}}{w_{d}^{2}}arctan\left(\frac{w_{d}}{z}\right). \end{array}$$

As $\tilde{\gamma }\left(z\right)$ is a multi-value function on the complex plane [32], it is complicated to obtained a closed form of the VAF. Nevertheless, we can analyze the behavior of the VAF for asymptotic time. When $w_{d}$ is equal to a finite value in Equation (Equation 6), the characteristic equation, $z+\tilde{\gamma }\left(z\right)=0$, has a pair of pure complex roots. Formally, the exact expression of the VAF for asymptotic time can be obtained, which is given by
(8)$$\begin{array}{c} C_{V}\left(t\right)=2c_{0}cos\left(y_{0}t\right),\;for\;asymptotic\;time \end{array}$$
(9)$$\begin{array}{c} y_{0}-\frac{3\gamma_{0}^{2}}{2w_{d}^{2}}Ln|\frac{y_{0}+w_{d}}{y_{0}-w_{d}}|=0;\;c_{0}={\left[1-\frac{3\gamma_{0}^{2}}{w_{d}^{2}}\frac{w_{d}}{\left(w_{d}^{2}-y_{0}^{2}\right)}\right]}^{-1};\; \end{array}$$
where $\pm iy_{0}$ are two pure imaginary roots of the characteristic equation. The coefficient $c_{0}$ denotes the residues of the imaginary roots. For example, for $w_{d}=0.5$, $y_{0}=2.47$, and $c_{0}=0.494$; for $w_{d}=1.5$, $y_{0}=1.74$, and $c_{0}=0.280$; and for $w_{d}=2.0$, $y_{0}=2.02$, and $c_{0}=0.0505$. In Figure 1, we plot numerical and analytical results. For asymptotic time, analytical results are in good agreement with numerical results by the numerical integration of Equation (Equation 5). From the Khinchin theorem [33], which states that if the autocorrelation function $C_{A}\left(t\right)$ of a variable *A* satisfies $C_{A}\left(t\to \infty \right)=0$, then *A* is an ergodic variable; ergodicity breakdown in the classical and quantum systems is observed because of frequencies cut-off. Notably, the ergodicity is exhibited when $w_{d}\to \infty$ as $y_{0}\to 0$ and $C_{V}\left(t\to \infty \right)=0$.

## 3. Two Effects on Escape Dynamics of Classical and Quantum Open Systems

### 3.1. Effect of $w_{d}$

We use the second-order Runge–Kutta algorithm [34,35,36] with a small time-step, h = 0.005, to study numerically how the steady escape rate of a self-oscillated system depends on $w_{d}$ in classical and quantum regimes, respectively. A type of metastable potential profile is chosen to be
(10)$$\begin{array}{c} U\left(X\right)=\left\{\begin{array}{cc} \frac{1}{2}w_{a}^{2}X^{2}, & \;X\leq X_{C}\\ U-\frac{1}{2}w_{b}^{2}{\left(X-X_{b}\right)}^{2}, & \;X>X_{C} \end{array}\right., \end{array}$$
where $X_{a}=0$ and $X_{b}$ are the coordinates of the potential well bottom and saddle point, respectively. *U* is the well depth. $X_{c}$ is the linking point of two smooth quadratic potentials. $w_{a}$ and $w_{b}$ are the frequencies of a harmonic potential and an inverse harmonic one, respectively. The time-dependent escape rate of the particle is determined by $r\left(t\right)=-\frac{1}{N(t)}\frac{\Delta N(t)}{\Delta t}$ [36], where N(t) denotes the number of particles that have not arrived the exit. We chose the exit, $X_{e}=7.5$, which is larger than the saddle point $X_{b}\approx 1.4$ with the choice of $w_{a}=w_{b}=2.0$ in our simulation. ΔN(t) is the number of particles crossing the exit first time during the period of t→t+Δt. As the exit is chosen far enough, which denotes that the particle cannot come back across the saddle point, it is removed once crossing the exit. For easy statistical analysis, we chose a finite time interval $t_{s}=0.3$. Moreover, the time-dependent escape rate during the interval $t\to t+t_{s}$ is given by [37]
(11)$$\begin{array}{c} \begin{array}{cc} r(t) & =\frac{1}{t_{s}}\int_{t}^{t+t_{s}}\left(-\frac{1}{N(t)}\frac{dN(t)}{dt}\right)dt\\ & =\frac{1}{t_{s}}\left[ln\left(N\left(t\right)\right)-ln\left(N\left(t+t_{s}\right)\right)\right]. \end{array} \end{array}$$
Initially, in the classical regime, the velocity and coordinate obey a Gaussian distribution with zero-mean and variance, $\langle V^{2}\rangle =\frac{k_{B}T}{M}$ and $\langle X^{2}\rangle =\frac{k_{B}T}{Mw_{a}^{2}}$. In the quantum regime, the velocity and coordinate obey a Gaussian distribution with zero-mean and variance [25], $\langle V^{2}\rangle =\frac{\hbar w_{a}}{2}coth\left(\frac{\hbar w_{a}}{2k_{B}T}\right)$ and $\langle X^{2}\rangle =\frac{\hbar }{2w_{a}}coth\left(\frac{\hbar w_{a}}{2k_{B}T}\right)$.

Panels (a,b) in Figure 2 show the time-dependent escape rate with various $w_{d}$ in the classical and quantum regime, respectively. In both regimes, it is obvious that the time-dependent escape rate arrives to oscillate around a constant after a period of time. The transient stage lasts approximately $t_{1}=10$ for different values of $w_{d}$. Thus, we get the steady escape rate, $r_{st}$, by time-averaging over r(t), which is given by $r_{st}=\frac{1}{t_{2}-t_{1}}\int_{t_{1}}^{t_{2}}r\left(t\right)dt$, where we choose $t_{1}=10$ and $t_{2}=25$. Moreover, panel (c) in Figure 2 shows that the steady escape rate depends non-monotonically on $w_{d}$, which means that there is an optimal cut-off frequency that makes the steady escape rate maximal.

The escape behavior of a self-oscillated system is analyzed by two timescales: the correlation time of fluctuations, $\tau_{c}$, and escape time, $\tau_{e}∼\frac{1}{w_{a}}exp\left(\frac{U}{k_{B}T}\right)$ [28], and a crossover between weak and strong friction regime [29]. For the memory kernel given by Equation (Equation 6), $\tau_{c}∼\frac{1}{w_{d}}$. The zero frequency friction is given by $\gamma_{eff}=\int_{0}^{\infty }\gamma \left(t\right)dt=\frac{3}{2}\frac{\pi \gamma_{0}^{2}}{w_{d}^{2}}=\xi_{0}$ [38]. On the one hand, self-oscillation is reported in both classical and quantum open systems. When $w_{d}$ is low, $\tau_{c}$>$\tau_{e}$, which means that escape process is greatly under the influence of self-oscillation of the system. The dynamics of the system is a non-Markovian process. In other words, the system has a strong memory of its initial states, which also means that the system is in the strong friction regime ($\gamma_{eff}\gg 2w_{a}$). [39] From Kramers’ theory [18], the steady escape rate can be given by $r_{st}\to \frac{w_{a}w_{b}}{2\pi \xi_{0}}\exp\left(-\frac{U}{K_{B}T}\right)$. When $w_{d}$ is moderate, the transition state theory and the Grote–Hynes formula [28,29] can be used to analyze the change of the escape rate $r_{st}$, which is given by $r_{st}=\frac{u}{w_{b}}\frac{w_{a}}{2\pi }\exp\left(-\frac{U}{K_{B}T}\right),$ where the real positive-valued quantity *u* can be determined by $u^{2}+u\hat{\gamma }\left(u\right)-w_{b}^{2}=0$, where $\hat{\gamma }\left(z\right)$ is given by Equation (Equation 7). Under the condition that $w_{d}$ is moderate so that the result is valid, the steady escape rate increases as $w_{d}$ increases by solving equation of *u* numerically.

On the other hand, when $w_{d}$ is high enough so that the system momentum varies sufficiently slowly over times of the order of $\tau_{c}$, the dissipative memory kernel can be approximated by a δ function, i.e., $\gamma \left(t\right)≃2\xi_{0}\delta \left(t\right);\xi_{0}=\frac{3\pi \gamma_{0}^{2}}{2w_{d}^{2}}$. The dynamics of the system is Markovian process and the friction acting on the system is weak. From Kramers’ theory [18], the steady escape rate can be given by $r_{st}\to \xi_{0}\frac{U}{k_{B}T}\exp\left(-\frac{U}{K_{B}T}\right)$. Therefore, as the high value of $w_{d}$ increases, it is easy to demonstrate that the steady escape rate decreases gradually to zero. As a result, the steady escape rate depends non-monotonically on $w_{d}$ for a crossover between weak and strong friction regime.

Comparing the steady escape rate of a quantum open system to a classical open system, it is no difficult to find that quantum fluctuation enhances the steady escape rate. It may be easy to verify that the QGLE in c numbers reduces to the GLE in the thermal limit $\hbar w_{i}\ll k_{B}T$ [26], where $\left\{w_{i}\right\}$ are the vibratory modes of oscillators in the heat bath. Therefore, the steady escape rate of quantum open system reduces classical open system as $w_{d}$ decreases gradually when the thermal limit holds.

### 3.2. Effect of *N*

Many examples for non-Markovian ergodicity breaking in a finite-size bath [13,19,24,31] are with a non-vanishing VAF of a force-free particle being non-stationary. A finite bath with limited resources, namely, a finite number of degrees of freedom, leads naturally to a cut-off for the density of the bath, which means many frequencies lack in the presence of a realistic heat bath. We start with (2N+2) Hamilton equations of a classical open system to investigate the effect of a finite number of oscillators on the steady escape rate by varying *N* from small to large. The equations of motion are given by
(12)$$\begin{array}{c} \left\{\begin{array}{cc} \dot{X} & =\frac{\partial H}{\partial P}=\frac{P}{M},\\ \dot{P} & =-\frac{\partial H}{\partial X}\\ & =-U^{\prime }\left(X\right)+\sum_{i=1}^{N}c_{i}\left(x_{i}-\frac{c_{i}}{m_{i}w_{i}^{2}}X\left(t\right)\right),\\ \dot{x_{i}} & =\frac{\partial H}{\partial p_{i}}=\frac{p_{i}}{m_{i}},\\ \dot{p_{i}} & =-\frac{\partial H}{\partial x_{i}}=-m_{i}w_{i}^{2}x_{i}+c_{i}X\left(t\right), \end{array}\right.\;. \end{array}$$

We use fourth-order Runge–Kutta algorithm with a small time-step, h=0.005, to study numerically how the steady escape rate depends on *N*. We consider a statistical average over 200,000 test particles. Each of the test particles is coupled to a bath composed by N-independent harmonic oscillators. The initial velocity distribution of test particles is assumed to be Gaussian with zero-mean and variance, $\langle V^{2}\rangle$=1.0. We suppose that oscillators in the bath are in thermal equilibrium with $K_{B}T$=1.0 at the initial time, where $k_{B}$ is the Boltzmann constant and *T* is the temperature of the bath. Moreover, the vibrational frequencies of oscillators are chosen randomly from a frequency distribution of Debye type. The frequency distribution function is $g\left(w\right)=\frac{3}{w_{d}^{3}}w^{2}$ for $w<w_{d}$ and g(w)=0 for $w>w_{d}$, where $w_{d}$ is a cut-off frequency. As our interest is to study escape dynamics by changing *N*, we fix values of $w_{d}$.

Panels (a,b) in Figure 3 show the time-dependent escape rate with various *N* when we fix $w_{d}=2.5$ and $w_{d}=3.0$ in a frequency distribution of the Debye type, respectively. After a period of time, the time-dependent escape rate starts to oscillate around a constant. Using the same methods, we obtain the dependence of the steady escape rate on *N* in panel (c) in Figure 3, which shows that the steady escape rate increases as the number of *N* increases gradually.

In our approach, due to the finite number of oscillators, the spectral density is always structure for low values of N. By plotting the frequency distribution g(w) for different values of *N* varying from small to large in Figure 4, it is clear that many frequencies are missing in the interval of interest when N=10, N=30, and N=120. Namely, ergodicity breaks when the value of *N* is low and the dynamics of system is non-Markovian process, which means that the friction exerting on the particle is strong. As *N* increases gradually, the friction becomes weak. In the limit of N→∞, ergodicity recovers with a high $w_{d}$ and the dynamics of the system becomes a Markovian process. Therefore, during the dynamics of the system going from a non-Markovian to Markovian process, the steady escape rate increases gradually.

## 4. Conclusions

We have analyzed ergodicity breakdown in classical and quantum open systems described, respectively, by a GLE and a QGLE in c numbers, both analytically and numerically, which is caused by a harmonic oscillator bath of Debye type. The VAF has been shown to oscillate in cosine function for asymptotic time. Escape of a self-oscillated open system from a metastable potential has shown interesting phenomena. On the one hand, the steady escape rate depends non-monotonically on $w_{d}$ because of the influence of self-oscillation, which has been analyzed by considering two timescales, $\tau_{c}$ and $\tau_{e}$. Comparing classical and quantum results, quantum fluctuation enhances the steady escape rate. On the other hand, the effect of a small number of oscillators in heat bath has been shown to decrease the steady escape rate comparing with large N.

The effect of self-oscillation on escape dynamics of open systems can be presented more intuitive through the present work. We believe that the present study will provide useful information about the study of the escape processes of open systems. Thus, some surprising findings may be revealed.

## Figures and Tables

**Figure 1 entropy-22-00839-f001:**
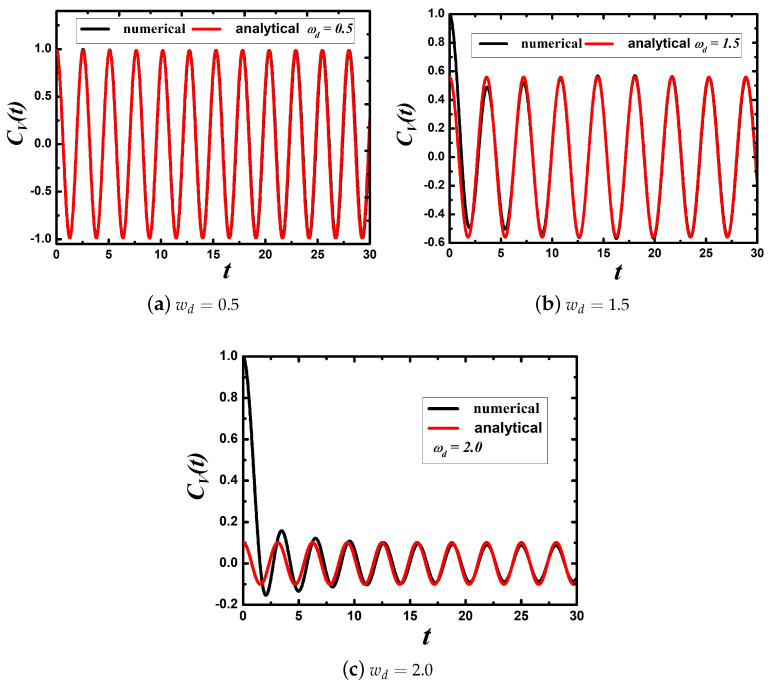
The VAF of a damped free particle with cut-off frequencies $w_{d}=0.5$, $w_{d}=1.5$, and $w_{d}=2.0$ in order from left to right in the figure. The black solid lines were obtained from the numerical integration of Equation (Equation 5) with Equation (Equation 6). The red lines are analytical results obtained from Equations (Equation 8) and (Equation 9).

**Figure 2 entropy-22-00839-f002:**
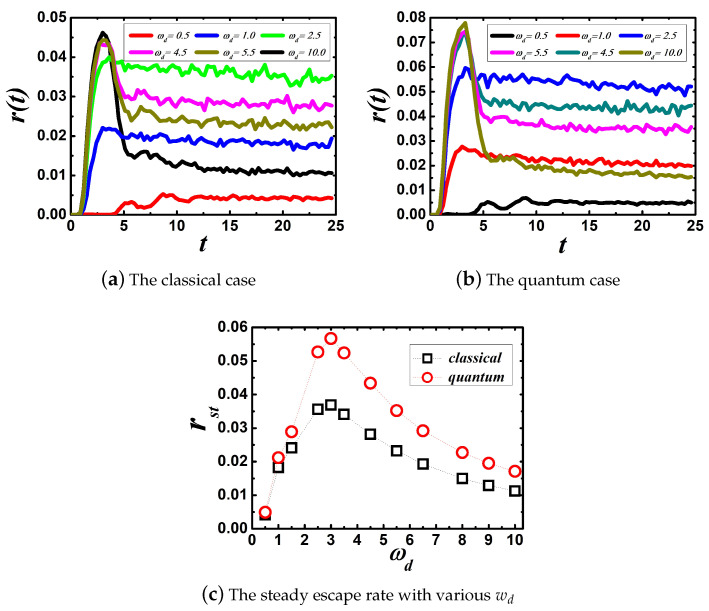
Panels (**a**,**b**), respectively, denote the time-dependent escape rate with different values of $w_{d}$ in the classical and quantum case. (**c**) The steady escape rate for different values of $w_{d}$. The parameters used are $k_{B}=1.0$, T=1.0, M=1.0, and $\gamma_{0}=1.0$. The parameters of the metastable potential are $w_{a}=2.0$, $w_{b}=2.0$, and U=2.0. All curves in panels (**a**,**b**) were plotted from results obtained by respectively simulating Equations (Equation 2) and (Equation 4) with 250,000 test particles. The black open squares in panel (**c**) denote the classical case and the red open circles the quantum case.

**Figure 3 entropy-22-00839-f003:**
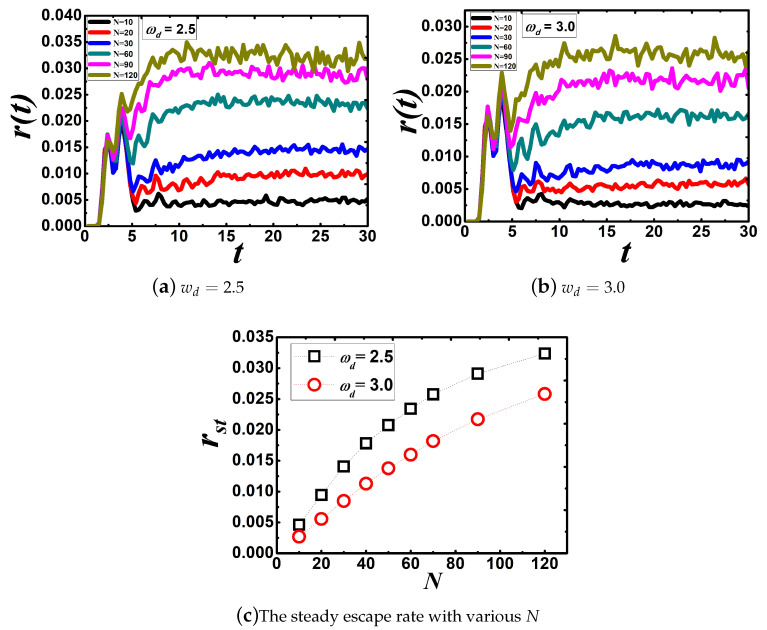
Panels (**a**,**b**) denote the time-dependent escape rate with different values of *N* with $w_{d}=2.5$ and $w_{d}=3.0$, respectively. (**c**) The steady escape rate for different values of *N*. The parameters used are $k_{B}=1.0$, T=1.0, $M=m_{i}=1.0$, and $c_{i}=0.1$. The parameters of the metastable potential are $w_{a}=2.0$, $w_{b}=2.0$, and U=2.0. All curves in panels (**a**,**b**) were plotted from results obtained by, respectively, simulating Equation (Equation 12) with 200,000 test particles. The black open squares in panel (**c**) denote $w_{d}=2.5$ and the red open circles $w_{d}=3.0$.

**Figure 4 entropy-22-00839-f004:**
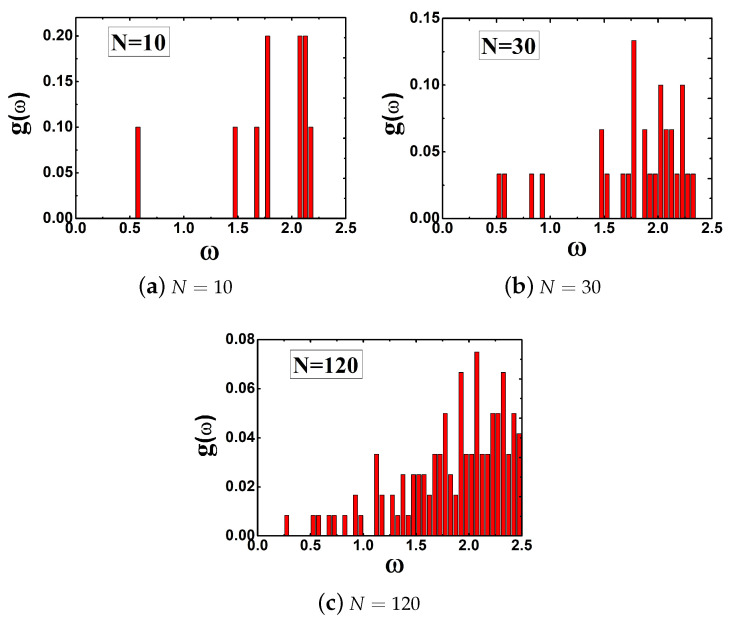
Frequency distribution for different values of N and $w_{d}$ = 2.5.

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
