# Peer review of "Effect of Self-Oscillation on Escape Dynamics of Classical and Quantum Open Systems"

_entropy, 2020, doi:10.3390/e22080839_

Round 1

Reviewer 1 Report

The manuscript entitled “Effect of Self-oscillation on Escape Dynamics of Classical and Quantum Open Systems” reports an interesting study on the effect of self-oscillations to the escape dynamics. A system-environment decomposition is employed either in classical or in quantum open systems. The memory kernel in the corresponding generalized Langevin equation is modeled by the Debye spectrum of a harmonic oscillator bath.

  1. Let us start from the Eq. (1), which defines mechanically the problem. Setting the coupling constants zero, the total Hamiltonian H reduces to Ho for the independent system and environment. The remaining interaction potential H-Ho must depend, however, solely on the distance between the particles, which is obviously not the case here. A detailed physical derivation of the Langevin dynamics from the rigorous interaction potentials is given by Tsekov and Ruckenstein, Chem. Phys. 100 (1994) 1450. It is shown that the ‘external’ potential U of Li and Bao contains parts of the interaction potential, which could affect essentially the considered escape dynamics.
  2. The exact memory kernel is inversely proportional to the cube of the Debye frequency, not to the square as postulate in Eq. (6), which compromises the conclusions in part 3.1.
  3. Tsekov and Ruckenstein considered in their paper exactly the potentials (10). They showed that the relevant friction coefficients depend strongly on the frequencies wa and wb, which contradicts to the model used by Li and Bao.

Author Response

Thanks for your valuable comments and guidance on the work. Please see the attachment.

Reviewer 2 Report

The authors analyze the quantum and thermal tunneling dynamics using a simple one-dimensional model, a system coupled to a bath of many harmonic oscillators. In particular, they examine the degree of Ergodicity by changing the cutoff frequency and the total number of oscillators of the bath. Applying numerical simulation with some analytical approximations, they derive the result: the larger the cutoff and the number, the larger the Ergodicity. Quite reasonable.

Although it is fun to examine the well-established problem by numerical examinations, the study seems to be insufficient to derive any novel results at the present level of numerical calculations. Therefore, the referee recommends a vast revision of the numerical calculations as well as the manuscript, and to derive more new physics before the reconsideration of the publication of this manuscript. The detail is as follows.

  1. What is the message of fig.1? The referee cannot get any relevance of this figure. If the author wants to show the relevance/validity of the analytical formula, a display of the whole range result would be needed; only showing the partial result t>5 is weird. Furthermore, the discrete display of the analytical result is weirder; Eq.8-9 should be continuously drawn in the graph.
  2. 1. If you want to compare the \omega_d dependence, the values chosen 1.75 and 2.0 are too close with each other. The readers would like to know vast tendencies in the logarithmic diagram, although all the graphs in this manuscript are on a linear scale.
  3. The choice of the parameter \omega_a=2 and \omega_b=2 are close to \omega_d. It might have simply induced the resonance which is far from Ergodicity, from the beginning. It would be more interesting to examine the dependence of the value \omega_b for tunneling.
  4. L112-l113 There is an abstract statement: `Since the exit is chosen far enough, it is removed once the particle crosses the exit.` Would it be better to specify the actual procedure or the boundary conditions?
  5. L122 The values t1=10, t2=25 seem to be chosen artificially. Naturally, t1 should be 0 and t2 infinity or the maximum time of the simulation.  
  6. l140 There is a statement `quantum fluctuation enhances the steady escape rate.`. On the other hand, it is often claimed that dissipation reduces the quantum tunneling rate because the energy loss destroys the quantum coherence. What is the relation to it? Is it consistent with each other?
  7. L156 and Fig3. Despite the statement, `we choose high values of w_d `, the authors actually choose w_d=2.5,3.0. The readers would like to know the case, for example, w_d=10 which was possible in Fig.2.
  8. What is the message of fig.4? It is quite confusing. Do they represent the frequency distribution of the Debye type with a limited number of sampling?
  9. l179 The statement ` On the other hand, self-oscillation caused by a finite number of oscillators in a heat bath has been proved to decrease the steady escape rate` seems to be inconsistent with the fact. Actually, the two facts, a) from fig 2(c), larger \omega_d reduces the steady escape rate r, and b) from fig 1, larger \omega_d reduces the amplitude of the fluctuation, suggest that the larger fluctuation enhances r.

Author Response

(The authors gave the same response as above.)

Reviewer 3 Report

The authors of the manuscript entitled “Effect of Self-oscillation on Escape Dynamics of Classical and Quantum Open Systems” (here after: the Authors) present a numerical study of a model escape problem, considering both the classical or quantum cases. The bath is modelled by a collection of oscillators a la Caldeira-Leggett and the authors investigate the effect of: 1) A finite frequency cut-off (Debye frequency) and 2) a finite number of oscillators on the escape dynamics.

I have some major concern on both the content and the methods of the manuscript. I would like the authors to address the points below before suggesting the manuscript for publication.

  • The authors should consider the work of Stella et al. Phys. Rev. B 89, 134303 (2014) in which a very similar problem has been studied in some detail, especially in their Sec III B. There it is suggested that a persistent oscillation (or self-oscillation according to the Authors) develops if the Debye frequency, say w_d, is smaller than any characteristic frequency of the system. This is particularly relevant in the case of the piecewise parabolic potential escape model that the Authors consider in the Sec. 3 of their manuscript. According to their escape model, there are two characteristic frequencies, w_a and w_b. I strongly suggest to discuss their results comparing w_a and w_b against w_d. Even in the thermodynamic limit, N to infinity, the system is poorly damped if w_a and w_b are much larger than w_d.
  • Although the Authors discuss their results using the Kramers’ theory and mention the comprehensive review by Haeggi et al. (their Ref. [28]), they do not provide many details. Do they have in mind the weak friction regime of Kramers’ theory? In the case, why? A non-monotonic behaviour of the transition rate is most often related to a cross-over between weak and strong friction regime (the Kramers’ turnover problem). I strongly suggest the Authors to discuss this point. In the discussion the Grote-Hynes formula may also help analyse their results.
  • The details of the simulation of the VAF in Sec. 3.1 of the manuscripts should be included in the manuscripts. Which equations of motion have been integrated? How many oscillators? Please present all the parameters of the Hamiltonian, including the oscillators’ bath parameters.
  • Can the Authors please state how they integrate their Eq. (5)? In particular, it is not clear how the frictional integral has been treated numerically.
  • Can the Authors please state clearly the regularity requirements (continuity of derivatives) of their piecewise parabolic potential defined in Eq. (10)?
  • Line 152, can the Authors please comment about their choice of <V^2>? It seems pretty arbitrary.
  • For the results of Sec. 3.2 (see their Fig. 3.), it seems that the Authors only consider the case w_a, w_b < w_d. Did they also simulate the w_a, w_b > w_d? The later would be more relevant for the observation of self-oscillation effect on the escape, as they claim in the title of the manuscript.
  • Line 146, the Authors should rephrase the sentence containing “caused by many frequencies lacking” more clearly. I think they refer to the frequencies missing in their random sampling, see their Fig. 4. In fact, I do not think the Authors clearly state that their frequency sampling is random. In any case, the Authors should consider an average over several instances of the random sampling of the bath frequencies. The results for a single instance may be not statistically significant. Please also refer to Stella et al. Phys. Rev. B 89, 134303 (2014) for an efficient non-random choice of the oscillators’ bath sampling
  • Although I understand the results of the Authors regarding the convergence to the thermodynamic limit, N to infinity, I think they should state this point more clearly. The point is that the system is not damped if there are no bath oscillators close to resonance. The discussion can be more soundly based on the discussion of the eigenvectors and eigenvalues of the equivalent dynamic systems, see e.g., Chem. Phys. 116, 2516 (2002). Please note that in Stella et al. Phys. Rev. B 89, 134303 (2014) as many as 500 oscillators have been used to accurately model a persistent oscillation (or self-oscillation) as it is not easy to converge to the thermodynamic limit.
  • Line 111, how is the exit condition defined and numerically detected in practice?
  • Line 112, it is not clear what “far enough” means in the context? Is it a comment about the position of the saddle or its energy?
  • In their conclusions, the Authors mention “strong memory” without quantifying its “strength”. Can the Authors please define the “strength” of the memory in terms of the parameters of the oscillators’ bath?
  • A minor point about the writing style. Please do not start a sentence with “And”. This is not a standard English usage of a conjunction.

Author Response

(The authors gave the same response as above.)

Round 2

Reviewer 1 Report

In the cited 1973 paper Zwanzig did a mistake in Eq. (25): the Debye frequency should be on third power in order the frequency distribution to be normalized. Li and Bao used the correct expression (row 168) but took mechanically the wrong Zwanzig result from his Eq. (28). Moreover, in the Zwanzig Eq. (23) the frequency appears on second power in the denominator! Li and Bao claimed at row 78 that the frequency is on first power but their integral does not result in Eq. (6).

The parameter gamma_0 = 1 is too rough approximation. According to Tsekov and Ruckenstein, gamma_0 is proportional to the square of wa and wb, respectively.

Author Response

Thanks for your critical feedback. We are very sorry that our first revised manuscript didn’t achieve your requirements. Please see the attachment.

Reviewer 2 Report

The authors sincerely consider all the comments and recommendation of the referee and mostly appropriately rewritten their manuscript. Therefore, the present referee recommends the publication of this manuscript to the journal. 

Author Response

Thanks for the reviewer’s positive evaluation on our submission.

Reviewer 3 Report

The authors of the manuscript entitled “Effect of Self-oscillation on Escape Dynamics of Classical and Quantum Open Systems” have very carefully addressed all my points. I am very happy to suggest their manuscript for publication.

I would also like to reassure the Authors, and especially the first one, that there is no need to be sorry about any unclear and undetailed statement. The Reviewer just wanted to clarify some point to improve the readability of the manuscript.

Author Response

(The authors gave the same response as above.)
